# The Genetics of Primary Familial Brain Calcification: A Literature Review

**DOI:** 10.3390/ijms241310886

**Published:** 2023-06-29

**Authors:** Shih-Ying Chen, Chen-Jui Ho, Yan-Ting Lu, Chih-Hsiang Lin, Min-Yu Lan, Meng-Han Tsai

**Affiliations:** 1Department of Neurology, Kaohsiung Chang Gung Memorial Hospital, College of Medicine, Chang Gung University, Kaohsiung 833401, Taiwan; k8911292001@cgmh.org.tw (S.-Y.C.);; 2Center for Parkinson’s Disease, Kaohsiung Chang Gung Memorial Hospital, College of Medicine, Chang Gung University, Kaohsiung 833401, Taiwan; 3Center for Mitochondrial Research and Medicine, Kaohsiung Chang Gung Memorial Hospital, College of Medicine, Chang Gung University, Kaohsiung 833401, Taiwan; 4School of Medicine, College of Medicine, Chang Gung University, Taoyuan 333323, Taiwan; 5Department of Medical Research, Kaohsiung Chang Gung Memorial Hospital, College of Medicine, Chang Gung University, Kaohsiung 833401, Taiwan; 6Genomics and Proteomics Core Laboratory, Kaohsiung Chang Gung Memorial Hospital, College of Medicine, Chang Gung University, Kaohsiung 833401, Taiwan

**Keywords:** primary familial brain calcification, Fahr’s disease, idiopathic basal ganglia calcification

## Abstract

Primary familial brain calcification (PFBC), also known as Fahr’s disease, is a rare inherited disorder characterized by bilateral calcification in the basal ganglia according to neuroimaging. Other brain regions, such as the thalamus, cerebellum, and subcortical white matter, can also be affected. Among the diverse clinical phenotypes, the most common manifestations are movement disorders, cognitive deficits, and psychiatric disturbances. Although patients with PFBC always exhibit brain calcification, nearly one-third of cases remain clinically asymptomatic. Due to advances in the genetics of PFBC, the diagnostic criteria of PFBC may need to be modified. Hitherto, seven genes have been associated with PFBC, including four dominant inherited genes (*SLC20A2*, *PDGFRB*, *PDGFB*, and *XPR1*) and three recessive inherited genes (*MYORG*, *JAM2*, and *CMPK2*). Nevertheless, around 50% of patients with PFBC do not have pathogenic variants in these genes, and further PFBC-associated genes are waiting to be identified. The function of currently known genes suggests that PFBC could be caused by the dysfunction of the neurovascular unit, the dysregulation of phosphate homeostasis, or mitochondrial dysfunction. An improved understanding of the underlying pathogenic mechanisms for PFBC may facilitate the development of novel therapies.

## 1. Introduction

Primary familial brain calcification (PFBC), previously known as Fahr’s disease or idiopathic basal ganglia calcification, is a rare inherited disorder characterized by bilateral calcification in the basal ganglia according to neuroimaging. Calcification may also affect other brain regions, such as the thalamus, cerebellum, and subcortical white matter. The clinical phenotypes are quite diverse, but the most common manifestations are movement disorders, cognitive deficits, and psychiatric disturbances. Although PFBC has nearly complete penetrance regarding neuroradiological findings, it has reduced penetrance with regard to clinical manifestations, with nearly one-third of cases being clinically asymptomatic [1,2,3]. The prevalence of PFBC is estimated to be 2.1–6.6 per 1000 persons [4,5], although it may be underestimated due to incomplete clinical penetrance and heterogenous presentations.

To date, two inheritance patterns have been found in PFBC patients. Heterozygous variants in *SLC20A2*, *PDGFRB*, *PDGFB*, and *XPR1* are responsible for autosomal dominant PFBC [6,7,8,9], while biallelic changes in *MYORG*, *JAM2*, and *CMPK2* are associated with autosomal recessive forms of the disease [10,11,12,13]. At present, around 50% of patients with PFBC do not have a pathogenic variant in the seven currently known genes [2], which indicates a more diverse genetic heterogeneity. The clinical findings and pathogenesis of PFBC have been described in details in recent reviews [14,15]. Therefore, this article focuses on the most pertinent knowledge of PFBC, emphasizing the recent advances in clinical genetics and summarizing the underlying mechanisms.

## 2. Clinical Presentations

Although PFBC can be found in all age groups, the onset usually occur between 40 and 60 years [3]. The age at onset among patients with *PDGFB* variants seems to be lower than that of those with *SLC20A2*, *XPR1*, and *MYORG* variants [3]. The clinical presentations of PFBC are quite diverse [16], with movement disorders, cognitive deficits, and psychiatric disturbances being most common ones [3]. Various types of movement disorders, including parkinsonism, tremors, dystonia, chorea, myoclonus, tics, and ataxia, have been reported, of which parkinsonism is the most frequent symptom [3,16]. In addition, paroxysmal kinesigenic dyskinesia has also been reported [17,18,19]. The cognitive deficits range from mild cognitive decline to dementia with frontal subcortical dysfunction [20], and memory and executive functions are the most commonly affected domains [21]. With regard to psychiatric disturbance, depression is the most commonly seen symptom, followed by psychosis and anxiety [3].

A wide range of speech disturbances, including dystonic, spastic, scanned, or slurred speech, can be presented in PFBC [16]; especially, a speech disturbance is the cardinal presentation of patients with *MYORG* variants [3,22]. The other neurological symptoms include various types of seizures, headaches/migraines, pyramidal signs, and stroke [21,23,24,25,26,27].

Interestingly, due to advances in neuroimaging and sequencing technology, around one-third of patients with brain calcification and pathogenic variants in PFBC genes remain clinically asymptomatic during their lifetime [1,2,3]. The penetrance varies among different PFBC genes; *PDGFB*, *MYORG*, and *JAM2* have the highest clinical penetrance (more than 85%), followed by *XPR1* and *SLC20A2* (70% and 60%, respectively). In contrast, *PDGFRB* has the lowest clinical penetrance (46%) [3]. The clinical variability and reduced penetrance may be influenced by other yet unidentified genetic modifiers [28].

## 3. Imaging Features

Brain calcification is essential for the diagnosis of PFBC, and the best imaging tool to detect brain calcification is non-contrast computed tomography (CT). The bilateral basal ganglia, especially the internal globus pallidus, are preferentially affected [29]. The other commonly involved brain areas include the thalamus, dentate nucleus, and subcortical white matter. Less frequently, calcifications can also be found in the internal capsule, cerebral and cerebellar cortex, and brainstem [3,16].

The pattern of brain calcification varies among different PFBC genes. Patients with biallelic variants in *MYORG* and *JAM2* tend to have more extensive calcified areas than patients with autosomal dominant PFBC genes do [3]. The distinctive features of *MYORG* related PFBC are calcification in the brainstem, especially in the pons, plus various degrees of cerebellar atrophy [22]. In contrast to *MYORG*, patients with *JAM2* have more severe and confluent bilateral parietal, temporal, and occipital cortical calcifications [12].

Of note, more restricted brain calcification, predominantly limited to the basal ganglia, can be found in monoallelic carriers of *MYORG*, *JAM2*, and *CMPK2* [13,30,31]. This suggests that the neuroradiological phenotype may be transmitted as a semi-dominant trait via heterozygous variants in autosomal recessive PFBC genes [15,30].

The magnetic resonance imaging (MRI) of the brain can provide better anatomic information than CT can; however, in PFBC, routine brain MRI is less sensitive and may lead to underdiagnosis [29]. New MRI techniques, such as susceptibility-weighted imaging (SWI), can exploit the magnetic susceptibility of various compounds, including intracranial calcification [32,33]. However, whether these new MRI technologies can replace CT as the first line diagnosis of PFBC remains to be thoroughly investigated.

## 4. Diagnostic Criteria

The current diagnostic criteria for PFBC were adapted from Moskowitz et al. [34], Ellie et al. [35], and Manyam et al. [36]. The diagnosis of PFBC is based on the presence of all four of the following criteria:Progressive neurological dysfunction, usually including movement and neuropsychiatric manifestations;The bilateral calcification of basal ganglia according to neuroimaging, with or without the involvement of other brain regions;The exclusion of other causes of calcification, such as metabolic problems, mitochondrial diseases, infectious, toxic, or traumatic causes;A positive family history of PFBC [29].

However, due to advances in the genetic research on PFBC in the past decade, some of the diagnostic criteria are no longer valid and may need to be modified. For example, not all patients present with clinical symptoms throughout their lifetime [3], and this may lead to the exclusion of some genetic and/or imaging-positive asymptomatic cases based on the current criteria. In addition, PFBC can be diagnosed in patients without a positive family history because affected relatives have unrecognized symptoms or are asymptomatic. Moreover, patients with de novo mutations or autosomal recessive inheritance can present with a negative family history [37]. Hence, the absence of a family history cannot completely rule out the possibility of PFBC, and the current diagnostic criteria may be suboptimal and may miss asymptomatic or non-familial cases. The future modification of the criteria to include genetic information may, thus, be required.

## 5. Differential Diagnosis

The calcification of the basal ganglia has also been identified in about 0.3–20% of individuals, especially among elderly people. Usually, age-related physiological calcification is not associated with clinical symptoms or a family history, and it may be misdiagnosed as PFBC [38,39,40]. To differentiate pathological basal ganglion calcification from frequently seen physiological calcification in normal aging, Nicolas et al. developed a visual grading system based on CT scans, called the total calcification score (TCS). Age-specific thresholds were determined according to the value of the 99th percentile of the TCS in different age categories among controls [21]. The authors concluded that the TCS is rarely affected by cerebral atrophy in the elderly, and therefore, it is a useful tool for the diagnosis of PFBC, with high inter-rater reliability [21]. However, the development of TCS predates the recent discovery of most of the PFBC genes. As a result, the use of the TCS for different forms of genetic PFBC requires further validation.

## 6. Genetics and Disease Mechanism

PFBC is genetically heterogeneous. To date, seven genes have been associated with PFBC, including four dominant genes and three recessive genes (Table 1).

### 6.1. SLC20A2

The first PFBC-causative gene, *SLC20A2*, was identified in 2012 [6]. It is located on chromosome 8 and encodes for type III sodium-dependent inorganic phosphate (Pi) transporter 2 (PiT2). PiT2 has 12 transmembrane domains, and this protein is responsible for the uptake of Pi into cells [2,6,16]. *SLC20A2* is expressed ubiquitously, but at a higher level in the brain, especially in the globus pallidus, thalamus, and cerebellum [41]. *SLC20A2* is the most common PFBC gene; heterozygous variants have been identified in more than 60% of genetically confirmed PFBC patients [3]. A missense change is the most common variant type, followed by frameshift, nonsense, and splice site variations, without obvious hotspots for pathogenic variants (Figure 1a) [3,6,7,17,18,21,23,37,48,49,50,51,52,53,54,55,56,57,58,59,60,61,62,63,64,65,66,67,68,69,70,71,72,73,74,75,76,77,78,79,80,81,82,83,84,85]. Functionally, both haploinsufficiency and dominant negative effects have been described; the loss of normal PiT2 function results in extracellular Pi accumulation and subsequent calcium phosphate formation [6,42]. Calcification around the blood vessels or within the vessel walls of involved brain regions has been demonstrated in *Slc20a2* homozygous knockout mice (*Slc20a2*^−/−^ mice) and in an autopsied *SLC20A2*-PFBC patient [86,87,88]. Calcification in *Slc20a2*^−/−^ mice was also found inside cells, mainly in the pericytes and astrocytes, which suggested the intracellular cytosolic initiation of calcification [87]. Moreover, increasing T-cell infiltration in the brain parenchyma was found in *Slc20a2*^−/−^ mice, which is positively associated with brain calcification and aging. Impaired blood–brain barrier (BBB) permeability with the enhancement of endocytosis and transcytosis was also demonstrated, which may be explained by the dysfunction of pericytes and astrocytes due to intracellular calcification [89]. In addition, PiT2 is known to be expressed in the apical membrane of choroid plexus epithelial cells in spiny dogfish and rats, suggesting that PiT2 plays an important role in actively transporting Pi from the cerebrospinal fluid (CSF) to the blood to maintain phosphate homeostasis in the CSF [90]. The level of Pi in CSF is significantly elevated in both *Slc20a2* homozygous knockout mice and PFBC patients with *SLC20A2* pathogenic variants [87,91,92]. In summary, PiT2 dysfunction can leads to a local increase in the extracellular and CSF Pi concentrations, which then trigger cell-mediated mineralization progression, ensuing calcification [86,87].

### 6.2. PDGFRB

The *PDGFRB* gene was identified in PFBC patients in 2013 [7], and it is located on chromosome 5 and encodes for platelet-derived growth factor receptor-β (PDGF-Rβ). The structure of PDGF-Rβ includes five extracellular immunoglobulin-like C2 type domains, a transmembrane domain, and a tyrosine kinase domain. PDGF-Rβ is a cell-surface tyrosine kinase receptor for members of the platelet-derived growth factor (PDGF) family, with a high affinity for homodimeric PDGF-B and PDGF-D. At the tissue level, the *PDGFRB* gene is highly expressed in the brain, especially in the basal ganglia and dentate nucleus of the cerebellum. At the cellular level, the *PDGFRB* gene is expressed in neurons, vascular smooth muscle cells (SCMs), and pericytes. The signal transduction of PDGF-Rβ and its ligand is essential for the proliferation and migration of vascular SCMs and pericytes, and subsequently, the angiogenesis and formation of the BBB [2,7,16]. Heterozygous variants in the *PDGFRB* gene have been identified in 5% of genetically confirmed PFBC patients [3]. The missense change is currently the only variant type to be identified in the *PDGFRB* gene. The variants tend to cluster in the tyrosine kinase domain (Figure 1b) [3,7,21,56,93,94], and they are likely to cause haploinsufficiency and affect the kinase function of the protein [43]. The functional loss of PDGF-Rβ may lead to pericytes dysfunction, which would impact BBB integrity and secondarily induce calcium depositions in the vessel walls or perivascular space. Several studies have shown that homodimeric PDGF-B can directly induce vascular SCM calcification via enhancing the expression of inorganic phosphate transporter 1 (PiT1) [95,96]. Hence, Nicolas et al. hypothesized that activating variants in *PDGFRB* may impact the PDGF-PiT1 pathway and cause vascular calcification [7]. Interestingly, PiT1 is encoded by the *SLC20A1* gene, which is one of two type III sodium-dependent Pi transporters. The other is PiT2 encoded by *SLC20A2* [7,94]. However, there is no further evidence supporting this hypothesis [14].

### 6.3. PDGFB

The *PDGFB* gene was also reported in 2013 [8], and it is located on chromosome 22 and encodes for the PDGF-B precursor protein. This precursor protein is cleaved at positions 81 and 191, and subsequently forms a homodimer through disulfide bonds, which is the main ligand of PDGF-Rβ [16]. The *PDGFB* gene is expressed in neurons and endothelial cells in the brain [8]. PDGF-B is a growth factor for mesenchymal cells and plays a crucial role in the proliferation and recruitment of pericytes and vascular SMCs. [44,45]. Heterozygous variants in the *PDGFB* gene have been identified in 12% of genetically confirmed PFBC patients [3]. The missense change is the most common variant type in the *PDGFB* gene, followed by nonsense, splice site, and extension variants. The variants cluster between protein positions 82 and 190, which are retained in the mature PDGF-B protein (Figure 1c) [3,8,17,56,59,69,82,97,98,99,100,101,102,103,104,105]. The same as *PDGFRB* does, *PDGFB* variants also cause haploinsufficiency, either by deleting critical parts of protein or disrupting normal protein function. The loss of normal PDGF-B function leads to BBB impairment via PDGF-Rβ, and then triggers the process of calcification [2,8].

### 6.4. XPR1

The *XPR1* gene was identified in 2015 [9], and it is located on chromosome 1 and encodes for xenotropic and polytropic retrovirus receptor 1 (*XPR1*). XPR contains eight transmembrane domains and an amino-terminal SPX domain [16]. This protein mediates Pi efflux from cells [2,9]. *XPR1* is expressed universally, and a high *XPR1* mRNA level has been demonstrated in mouse brains, especially in the cerebellum and striatum [46]. Heterozygous variants in the *XPR1* gene have been identified in 6% of genetically confirmed PFBC patients [3]. The missense change is the most common variant in *XPR1*-related PFBC patients, which tends to cluster in the SPX domain (Figure 1d) [3,9,19,53,56,106,107]. *XPR1* variants may cause haploinsufficiency, leading to the intracellular accumulation of Pi and formation of calcium phosphate [9]. Interestingly, mutual interactions between *XPR1* and *PDGFRB* were found in a recent immunoprecipitation study [46]. It is hypothesized that these two proteins may form a complex on the cell membrane, further suggesting that *PDGFRB* may be the upstream regulator of *XPR1* [46].

### 6.5. MYORG

In 2018, the first and most common autosomal recessive PFBC-causative gene, *MYORG*, was identified. *MYORG* is located on chromosome 9 and encodes for myogenesis-regulating glycosidase (*MYORG*). It contains a short cytoplasm domain at the N-terminal, a transmembrane domain, and a long luminal fragment with a glycosidase domain at the C-terminal. *MYORG* is a member of the glycosyl hydrolase 31 family, and its function is to regulate protein glycosylation. In the brain, the *MYORG* gene is highly expressed in the cerebellum, specifically in the endoplasmic reticulum of the astrocytes [10]. Biallelic variants in the *MYORG* gene have been identified in 13% of genetically confirmed PFBC patients [3]. The missense change is the most common variant type in the *MYORG* gene, followed by in-frame indels, nonsense, and frameshift variations. There is no obvious hotspot of pathogenic variants (Figure 1e) [3,10,22,24,25,26,27,30,108,109,110,111,112,113,114,115,116,117,118]. Pathogenic variants cause the loss of the glycosidase function of *MYORG*, which may lead to abnormal protein glycosylation and metabolic disturbance. It is believed that *MYORG* variants can induce astrocyte dysfunction, which then disturbs the association between astrocytes and pericytes, resulting in neurovascular unit (NVU) dysfunction and subsequently causing the formation of brain calcification [10]. However, the exact linkage between the loss of protein glycosylation and astrocyte dysfunction remains to be elucidated.

### 6.6. JAM2

In 2020, another autosomal recessive PFBC-causative gene, *JAM2*, was identified. *JAM2* is located on chromosome 21 and encodes for junctional-adhesion-molecule-2 (*JAM2*). The structure of *JAM2* includes two immunoglobulin-like domains (V-type and C2-type). *JAM2* is a member of the junctional adhesion molecule family, and it plays crucial roles in the regulation of cell polarity, endothelium permeability, leukocyte migration, and BBB function [11,12]. In the brain, *JAM2* is highly expressed in the caudate nuclei. At the cellular level, *JAM2* is specifically expressed in endothelial cells and astrocytes [12]. Biallelic variants in the *JAM2* gene have been identified in 2% of genetically confirmed PFBC patients [3]. The nonsense change is the most common variant type in the *JAM2* gene. Missense, frameshift, and structural variants have also been reported, without a mutation hotspot (Figure 1f) [3,11,12]. Variants of *JAM2* gene cause the loss of cell–cell adhesion and the dysfunction of the solute passage, which may contribute to the formation of brain calcification [11,12].

### 6.7. CMPK2

The *CMPK2* gene is the latest autosomal recessive PFBC-causative gene to be identified at the end of 2022 [13]. This gene is located on chromosome 2 and encodes for uridine monophosphate-cytidine monophosphate kinase 2 (UMP-*CMPK2*), which can be separated into N-terminal and C-terminal domains according to the sequence properties. *CMPK2* is a member of the nucleoside monophosphate kinase family and participates in the salvage pathway for the phosphorylation of dCMP, dUMP, CMP, and UMP in the mitochondria [47]. In the brain, the *CMPK2* gene is highly expressed in the hippocampus and cerebellum. At the cellular level, *CMPK2* is specifically expressed in neurons and vascular endothelial cells [13]. To date, biallelic variants in the *CMPK2* gene have only been reported in two PFBC families, which carry missense and start-codon loss variants (Figure 1g) [13]. The loss of UMP-*CMPK2* function leads to a reduction in mitochondrial genome DNA copy numbers, the downregulation of the expression of mitochondrial protein, the decrease in ATP production, and the disturbance of mitochondrial cristae morphology. The disturbance of mitochondrial function is believed to cause impairment of energy homeostasis and the upregulation of intracellular phosphate levels, subsequently triggering the development of brain calcification [13].

### 6.8. Possible Pathophysiological Mechanisms of PFBC

In conclusion, three potential mechanisms underlying the pathogenesis of PFBC have been proposed. The first mechanism is the disturbance of the homeostasis of phosphate, which is supported by the transporter function of *SLC20A2* and *XPR1*. The second mechanism is the disruption of the NVU. The NVU is composed of neurons, astrocytes, vascular SMCs, pericytes, and endothelial cells (Figure 2). Each element of the NVU interacts tightly with the others, which leads to an effective system used to regulate cerebral blood flow and maintain BBB integrity [119,120]. This is supported by the observation that *PDGFRB*, *PDGFB*, *MYORG*, and *JAM2* are highly expressed in the composite cells of the NVU. However, the exact mechanism of how NVU dysfunction causes brain calcification remains to be elucidated [10,11,12]. The third mechanism is the impairment of mitochondrial function. This hypothesis is supported by research on *CMPK2*, which plays a significant role in the salvage pathway in the mitochondria [13]. Interestingly, some connections between these three mechanisms have also been proposed. First, intracellular calcification in *Slc20a2*^−/−^ mice has been shown to be mainly distributed in NVU cells, such as the pericytes and astrocytes [87]. Second, the co-localization of *XPR1* and *PDGFRB* has been reported, and they may form a complex on the cell membrane [46]. Third, *CMPK2* is highly expressed in neurons and vascular endothelial cells, and the latter ones are also part of the NVU [13]. Taken together, the homeostasis of phosphate and maintaining mitochondrial function in NVU cells appear to play important roles in the pathogenesis of PFBC [12,13]. Further functional studies are needed to elucidate the detailed mechanisms.

### 6.9. Composition of Calcification in PFBC

The structures of calcium deposits have been reported to be similar in *Slc20a2*^−/−^ mice and PFBC patients, with the principal component being hydroxyapatite (Ca_10_[PO_4_]_6_[OH]_2_), and C, N, O, S, Fe, Zn, Al, and Mg were also observed [87,88]. The protein components in the calcification of hypomorphic mice with *PDGFB* (*PDGFB*^ret/ret^ mice) have also been identified. The most abundant proteins in calcification are known to regulate bone mineralization, including both a bone formation promotor, secreted protein acidic and rich in cysteine-like 1 (SPARCL1), and inhibitors, alpha 2-Heremans-Schmid glycoprotein (AHSG), matrix gla protein (MGP), and osteopontin (OPN) [121]. Moreover, the accumulation of reactive astrocytes and microglia around calcified area has been reported in both a PFBC patient and *PDGFB*^ret/ret^ mice [121,122]. Reactive astrocytes express proteins that function in oxidative damage and neurotoxic responses, and abnormal osteocyte markers, which indicate that reactive astrocytes may contribute to the formation of an osteogenic environment [14,122,123]. Reactive microglia acquire an osteoclast-like phenotype and may play a protective role in controlling brain calcification [123,124]. In summary, these findings suggest that the formation of calcification in PFBC may be a mineralization process rather than primary precipitation of calcium-phosphate products.

## 7. Treatment

Currently, there is no disease-modifying therapy for PFBC to reduce brain calcification or prevent its progression, and the treatment is mainly symptomatic. The discovery of PFBC-causative genes and clarifying the disease mechanisms in recent years has opened the door for therapeutic development. In one case series, biphosphonates, which modify the bone resorption and formation cycle, was shown to potentially stabilize the natural course of PFBC and showed symptomatic improvement among some younger patients. However, the study was small and included seven patients with heterogeneous PFBC, and thus, the therapeutic effect remains to be corroborated in larger clinical trials [125]. In a recent study, four *SLC20A2* variants (D28N, G120R, A227V, and C496Y) were investigated functionally, and the results showed that Pi transport activity in cells was abolished, except for A227V, which was partially maintained [55]. Interestingly, the presence of the A227V variant has been reported in healthy members of the PFBC family, suggesting that the partial preservation of Pi transport function may help suppress the onset of PFBC. Therefore, a treatment to upregulate the activity of PiT2 could be a potential direction for future development [55,126]. As mentioned, the severity of brain calcification seems to be associated with increased T-cell infiltration in the brain parenchyma. Fingolimod, a sphingosine-1-phosphate receptor modulator, is used to inhibit the circulation of peripheral T-cells, and it has been shown to reduce the trafficking of T cells into the central nervous system. In a recent animal study, brain calcification was significantly alleviated in *Slc20a2*^−/−^ mice after the intraperitoneal administration of fingolimod for three months. Thus, fingolimod may be a potential treatment for PFBC [89].

## 8. Conclusions

PFBC is a rare neurological disorder, with heterogeneous clinical presentations, neuroimaging findings, and clinical genetics. Seven genes have been linked to PFBC to date; however, around 50% of PFBC patients still have received no genetic diagnosis. Future investigations to identify novel PFBC-causative genes will likely fill the diagnostic gaps in these undiagnosed cases and shed more light on the pathogenesis of how this disease develops. The currently known PFBC genes play crucial roles in phosphate homeostasis and maintaining mitochondrial function, especially in the cells of the NVU, which may be targeted for the development of novel therapeutic interventions or gene therapy.

## Figures and Tables

**Figure 1 ijms-24-10886-f001:**
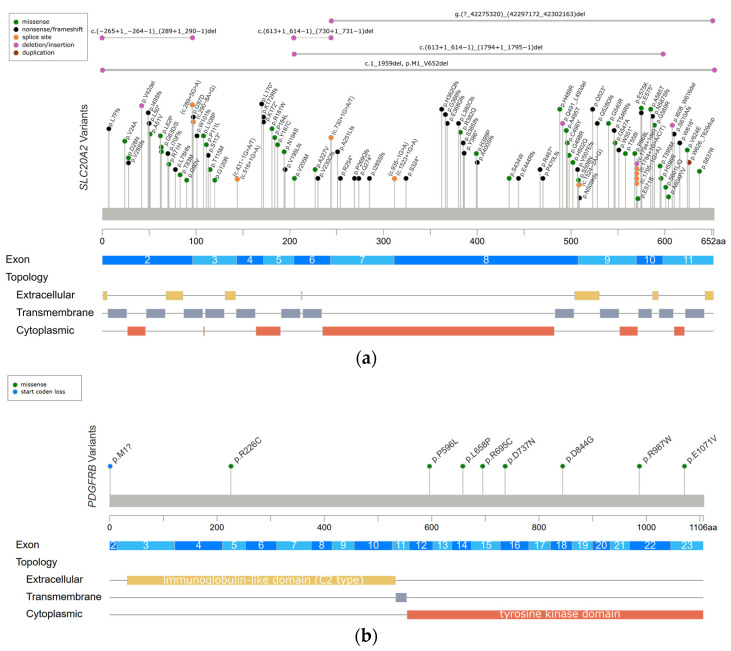
Reported variants in seven genes linked to PFBC along the protein sequence and their topologic protein models. (**a**) The reported variants in the *SLC20A2* gene and topology protein model of PiT2. (**b**) *PDGFRB* gene and PDGF-Rβ. (**c**) *PDGFB* gene. (**d**) *XPR1* gene and *XPR1*. (**e**) *MYORG* gene and *MYORG*. (**f**) *JAM2* gene and *JAM2*. (**g**) *CMPK2* gene. There is no obvious hotspot for *SLC20A2*, *MYORG*, and *JAM2* genes. The variants tend to cluster in the tyrosine kinase domain of the *PDGFRB* gene, the mature protein product between positions 82 and 190 of the *PDGFB* gene, and in the SPX domain of the *XPR1* gene. Meaning of symbols: *, stop codon; ?, unknown (a variant affecting the initiation codon cannot be predicted).

**Figure 2 ijms-24-10886-f002:**
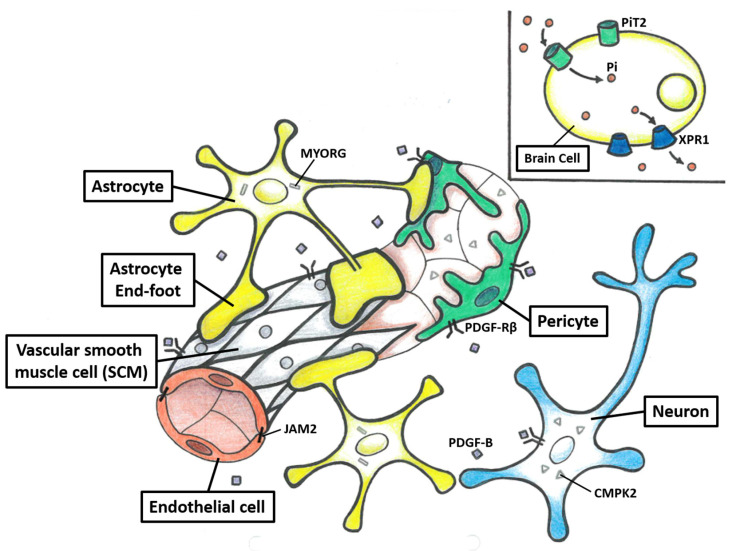
The structure of the neurovascular unit (NVU) and protein model of seven PFBC-causative genes. The NVU is composed of neurons, astrocytes, vascular SMCs, pericytes, and endothelial cells. Each element of the NVU interacts tightly with the other, which leads to an effective system used to regulate cerebral blood flow and maintain BBB integrity. *PDGFRB*, *PDGFB*, *MYORG*, and *JAM2* are highly expressed in the composite cells of the NVU. The neurons and endothelial cells secrete PDGF-B, which binds to PDGF-Rβ. The PDGF-Rβ is mainly encoded on the surface of neurons, vascular SCMs, and pericytes. *MYORG* is specifically encoded in the endoplasmic reticulum of the astrocytes. *JAM2* is mainly encoded in endovascular cells and astrocytes. *CMPK2* is highly encoded in neurons and endothelial cells. PiT2 and *XPR1*, which are highly encoded in the brain, mediate Pi uptake into cells and efflux from cells, respectively.

**Table 1 ijms-24-10886-t001:** Summary of PFBC-causative genes.

Gene	Locus	Mode of Inheritance	Protein	Expression	Function	Effect of Variant	Most Common Variant Type	References
*SLC20A2*	Chr 8	AD	Type III sodium-dependent inorganic phosphate transporter 2 (PiT2)	Ubiquitously, higher level in the brain	Uptake of inorganic phosphate (Pi) into cells	Loss of function	Missense	[3,6,41,42]
*PDGFRB*	Chr 5	AD	Platelet-derived growth factor receptor-β (*PDGFRB*)	Neurons, vascular smooth muscle cells (SCMs), pericytes in the brain	Cell-surface tyrosine kinase receptors for the PDGF family, especially for homodimeric PDGF-B and PDGF-D; Essential for angiogenesis and formation of the blood–brain barrier (BBB)	Loss of function	Only missense	[2,3,7,43]
*PDGFB*	Chr 22	AD	Platelet-derived growth factor subunit B (*PDGFB*)	Neurons and endothelial cells in the brain	Growth factors for mesenchymal cells; Crucial role in the proliferation and recruitment of pericytes and vascular SCMs	Loss of function	Missense	[3,8,44,45]
*XPR1*	Chr 1	AD	Xenotropic and polytropic retrovirus receptor 1 (*XPR1*)	Ubiquitously, higher level in the brain	Pi efflux from cells	Loss of function	Missense	[3,9,46]
*MYORG*	Chr 9	AR	Myogenesis regulating glycosidase (*MYORG*)	Endoplasmic reticulum of the astrocytes in the brain	Member of the glycosyl hydrolase 31 family; Regulate protein glycosylation	Loss of function	Missense	[3,10]
*JAM2*	Chr 21	AR	Junctional-adhesion-molecule-2 (*JAM2*)	Endothelial cells and astrocytes in the brain	Member of the junctional adhesion molecules family; Crucial role in the regulation of cell polarity, endothelium permeability, leukocyte migration, and BBB function	Loss of function	Nonsense	[3,11,12]
*CMPK2*	Chr 2	AR	Uridine monophosphate-cytidine monophosphate kinase 2 (UMP-*CMPK2*)	Neurons and endothelial cells in the brain	Takes part in the salvage pathway for phosphorylation of dCMP, dUMP, CMP, and UMP in the mitochondria	Loss of function	Missense and start-codon loss	[13,47]

Abbreviations: Chr: chromosome; AD: autosomal dominant; AR: autosomal recessive.

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
