# Peer review of "The Genetics of Primary Familial Brain Calcification: A Literature Review"

_ijms, 2023, doi:10.3390/ijms241310886_

Round 1

Reviewer 1 Report

This manuscript seems good enough as a educational review of "primary familial brain calcification.

>This manuscript is a review of the genetics of Primary Familial Brain Calcification.   >This is the review, which is educational, and described compactly and comprehensively.   >The references seems appropriate.

Author Response

We thank the reviewer for the positive comments.

Reviewer 2 Report

Manuscript ID ijms-2309457

I read with interest the review by Chen et al. 

Primary familial brain calcification (PFBC) is a certainly underdiagnosed condition with a large fraction of asymptomatic cases, and symptoms ranging from movement disorders to behavioral and psychiatric signs. It has often an enigmatic pathogenesis. To date, autosomal and recessive inheritance is described and at least 7 major mutated genes have been related to PFBC. Genetic etiology of PFBC can be classified into: - imbalance of P(i)homeostasis and - neurovascular unit (NVU) dysfunction. The topic is very emerging given that the comprehension of the pathophysiological mechanisms could of help to explain the common phenomenon of brain calcification and possible therapeutic approaches.

Comments to the AA:
Although the review by Chen S-Y et al. do summarize this knowledge, the work, in my opinion, remains at a merely descriptive level with a standard which could be adequate for an academic dissertation rather than for a scientific publication. Language and style are sufficient although not perfect.

The AA would deepen the clinical presentation (#2) and imaging features (#3) sections. I did not understand the paragraph title 6.8 Conclusion. Indeed, the AA would summarize these sentences in a separate paragraph as possible pathophysiological mechanisms of PFBC. I cannot see inspirational inputs that add extra-value beyond the needed literature search effort.

Bibliography should be updated. The AA should cite and discuss the following articles:

1) Carecchio M, Mainardi M, Bonato G. The clinical and genetic spectrum of primary familial brain calcification. J Neurol. 2023 Mar 2. doi: 10.1007/s00415-023-11650-0.

2)  Song T, Zhao Y, Wen G, Du J, Xu Q. A novel <i>MYORG</i> mutation causes primary familial brain calcification with migraine: Case report and literature review. Front Neurol. 2023 Feb 2;14:1110227.

3) Zhang Y, Ren Y, Zhang Y, Li Y, Xu C, Peng Z, Jia Y, Qiao S, Zhang Z, Shi L. T-cell infiltration in the central nervous system and their association with brain calcification in <i>Slc20a2</i>-deficient mice. Front Mol Neurosci. 2023 Jan 20;16:1073723. doi: 10.3389/fnmol.2023.1073723.

4) Xu X, Sun H, Luo J, Cheng X, Lv W, Luo W, Chen WJ, Xiong ZQ, Liu JY. The Pathology of Primary Familial Brain Calcification: Implications for Treatment. Neurosci Bull. 2022 Dec 5. doi: 10.1007/s12264-022-00980-0.

5) McKenna MC, Redmond J, Bradley D, Bede P. Teaching NeuroImage: Primary Familial Brain Calcification in <i>SLC20A2</i> Genotype. Neurology. 2022 Nov 29;99(22):1008-1009. doi: 10.1212/WNL.0000000000201343.

6)  Sadok SH, Borges-Medeiros RL, de Oliveira DF, Zatz M, de Oliveira JRM. Report of a young patient with brain calcifications with a novel homozygous MYORG variant. Gene. 2023 Apr 5;859:147213. doi: 10.1016/j.gene.2023.147213.

7) Maheshwari U, Huang SF, Sridhar S, Keller A. The Interplay Between Brain Vascular Calcification and Microglia. Front Aging Neurosci. 2022 Mar 2;14:848495. doi: 10.3389/fnagi.2022.848495.

In the present form, the article is not acceptable for publication.

Author Response

Dear Editor and Reviewers,

We thank the editor and reviewers for their time to help us improve our manuscript. We have addressed the reviewers’ suggestions point by point below:

  1. Although the review by Chen S-Y et al. do summarize this knowledge, the work, in my opinion, remains at a merely descriptive level with a standard which could be adequate for an academic dissertation rather than for a scientific publication. Language and style are sufficient although not perfect.

A: We thank the reviewer for the critical comments and suggestions to help us improve the literature review, we address the questions point by point below.

  1. The AA would deepen the clinical presentation (#2) and imaging features (#3) sections.

A: We thank the reviewer for the constructive suggestion. We added more information and revised the sentence in the paragraphs of clinical presentations (#2, p.2, line 52-53 and 66), and imaging features (#3, p.2, line 84-92).

  1. I did not understand the paragraph title 6.8 Conclusion. Indeed, the AA would summarize these sentences in a separate paragraph as possible pathophysiological mechanisms of PFBC. I cannot see inspirational inputs that add extra-value beyond the needed literature search effort.

A: We thank the reviewer for the suggestion. We changed the title 6.8 to “Possible pathophysiological mechanisms of PFBC” as suggested (p.9, line 304). Indeed, this is a literature review and we hope to summarize the recent advances in the field. The future discovery of new PFBC genes in unsolved cases may lead to a novel understanding of the disease mechanisms.

  1. Bibliography should be updated. The AA should cite and discuss the following articles:

1) Carecchio M, Mainardi M, Bonato G. The clinical and genetic spectrum of primary familial brain calcification. J Neurol. 2023 Mar 2. doi: 10.1007/s00415-023-11650-0.

2)  Song T, Zhao Y, Wen G, Du J, Xu Q. A novel <i>MYORG</i> mutation causes primary familial brain calcification with migraine: Case report and literature review. Front Neurol. 2023 Feb 2;14:1110227.

3) Zhang Y, Ren Y, Zhang Y, Li Y, Xu C, Peng Z, Jia Y, Qiao S, Zhang Z, Shi L. T-cell infiltration in the central nervous system and their association with brain calcification in <i>Slc20a2</i>-deficient mice. Front Mol Neurosci. 2023 Jan 20;16:1073723. doi: 10.3389/fnmol.2023.1073723.

4) Xu X, Sun H, Luo J, Cheng X, Lv W, Luo W, Chen WJ, Xiong ZQ, Liu JY. The Pathology of Primary Familial Brain Calcification: Implications for Treatment. Neurosci Bull. 2022 Dec 5. doi: 10.1007/s12264-022-00980-0.

5) McKenna MC, Redmond J, Bradley D, Bede P. Teaching NeuroImage: Primary Familial Brain Calcification in <i>SLC20A2</i> Genotype. Neurology. 2022 Nov 29;99(22):1008-1009. doi: 10.1212/WNL.0000000000201343.

6)  Sadok SH, Borges-Medeiros RL, de Oliveira DF, Zatz M, de Oliveira JRM. Report of a young patient with brain calcifications with a novel homozygous MYORG variant. Gene. 2023 Apr 5;859:147213. doi: 10.1016/j.gene.2023.147213.

7) Maheshwari U, Huang SF, Sridhar S, Keller A. The Interplay Between Brain Vascular Calcification and Microglia. Front Aging Neurosci. 2022 Mar 2;14:848495. doi: 10.3389/fnagi.2022.848495.

A: We thank the reviewer for the constructive suggestion. We updated the bibliography as suggested and revised the manuscript accordingly (p.5, line 159-163, p.10, line 339-355, and p.11, line 370-376), and updated the information in Figure 1(a) and 1(e) (p.7, line 283-284 and p.8, line 291-292).

Reviewer 3 Report

Overall, this is one well composed review. It provides detailed information on PFBC and especially its associated genes. It benefits the understanding  of pathogenesis of PFBC and exploring potential therapeutic.

1. The English writing should be improved;

2. In the Table 1, the refereances from which the information of the genes should be list out in one panel. 

Author Response

Dear Editor and Reviewers,

We thank the editor and reviewers for their time to help us improve our manuscript. We have addressed the reviewers’ suggestions point by point below:

  1. The English writing should be improved

A: The manuscript has been reviewed by a native English speaker, we hope that it is now more fluent and understandable.

  1. In the Table 1, the references from which the information of the genes should be listed out in one panel.

A: We thank the reviewer for the constructive suggestion. We add one panel for references in Table 1 (p.3-4, lines 139-140).

Round 2

Reviewer 2 Report

Ms. IJMS2309457

Although I appreciate the efforts made by the AA in addressing my points, I believe that the improvement is still not enough for acceptability of the manuscript. Unfortunately, as a researcher, reading this review does  not add anything new as compared to other similar review articles in the same field. For this very reason, I would like to give you a last opportunity in improving the quality of your work.

Author Response

We thank the reviewer for the constructive suggestion. The manuscript has been edited by a professional English editing service (please see the attachment for the certificate). We recognized that several excellent review articles existed on PFBC; we cited these publications in the Introduction clearly to refer our readers to them should the readers be interested to find out more information (#1, p.1-2, line 47-50). Compared to previous reviews, our manuscript focused more on clinical genetics and the details of different molecular mechanisms of PFBC. We also included the latest identified PFBC gene, CMPK2, which was published after several previous reviews. Moreover, we made summary figures of all the reported pathogenic variants of all 7 PFBC genes in detail, which has not existed in previous articles. We believe that the update and comprehensive figures are of interest to the researchers in the field as well as the readership of IJMS.

We hope that the reviewers and editors can see the merits of our efforts and accept for publication in IJMS.

Round 3

Reviewer 2 Report

Dear Authors,

I appreciate the efforts of the AAs in improving English language. I remain on my own view but I do not see major obstacles to publication. I can still see  minor english language and style errors. I recommend the AAs to revise again the text.

Author Response

The manuscript has been reviewed by a native English speaker again. We hope that it is now more fluent and understandable.